# Generative Graph Pattern Machine

**Zehong Wang[1], Zheyuan Zhang[1], Tianyi Ma[1], Chuxu Zhang[2], Yanfang Ye[1][†]**

[1]University of Notre Dame, [2]University of Connecticut

[†]Corresponding Author

`<zwang43,yye7>@nd.edu`

## Abstract

Graph neural networks (GNNs) have been predominantly driven by message-passing, where node representations are iteratively updated via local neighborhood aggregation. Despite their success, message-passing suffers from fundamental limitations—including constrained expressiveness, over-smoothing, over-squashing, and limited capacity to model long-range dependencies. These issues hinder scalability: increasing data size or model size often fails to yield improved performance. To this end, we explore pathways beyond message-passing and introduce **G**enerative **G**raph **P**attern **M**achine (**G²PM**), a generative Transformer pre-training framework for graphs. G²PM represents graph instances (nodes, edges, or entire graphs) as sequences of substructures, and employs generative pre-training over the sequences to learn generalizable and transferable representations. Empirically, G²PM demonstrates strong scalability: on the `ogbn-arxiv` benchmark, it continues to improve with model sizes up to 60M parameters, outperforming prior generative approaches that plateau at significantly smaller scales (e.g., 3M). In addition, we systematically analyze the model design space, highlighting key architectural choices that contribute to its scalability and generalization. Across diverse tasks—including node/link/graph classification, transfer learning, and cross-graph pretraining—G²PM consistently outperforms strong baselines, establishing a compelling foundation for scalable graph learning. The code and dataset are available at `https://github.com/Zehong-Wang/G2PM`.

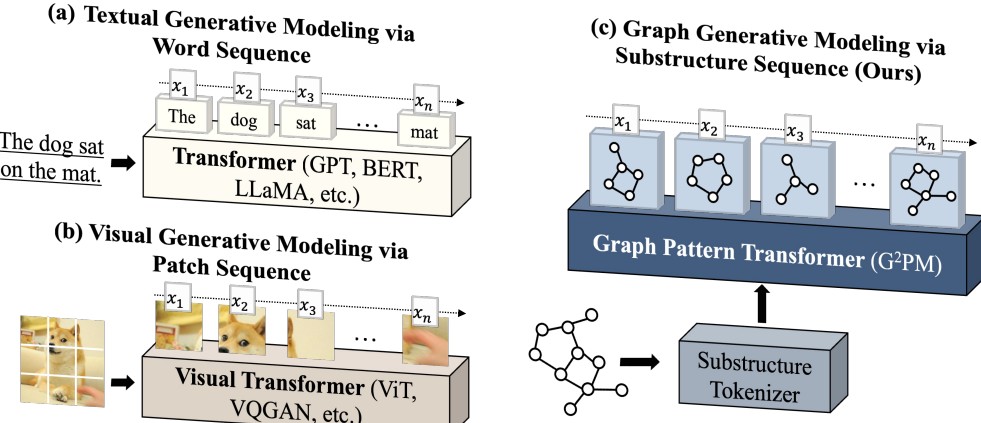

Figure 1: **Generative Transformer pre-training across modalities.** (a) *Textual modeling* tokenizes language into word sequences and generates next tokens or masked tokens. (b) *Visual modeling* slices images into patches and models generation in raster order. (c) *Graph modeling* (ours) tokenizes graph instances into substructure sequences using a random walk-based substructure tokenizer, and learns to generate via masked substructure modeling, going beyond message-passing.

39th Conference on Neural Information Processing Systems (NeurIPS 2025).

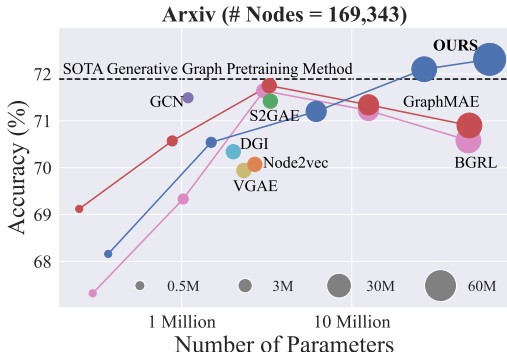

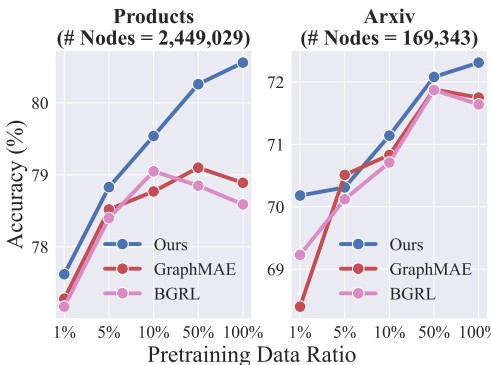

Figure 2: **Model scaling behavior** on the `ogbn-arxiv` with linear probing. $G^2PM$ consistently improves as model parameters increase, achieving 72.31% accuracy with 60M parameters—surpassing the current SOTA generative graph pre-training method [23]. In contrast, GraphMAE [22] and BGRL [53] exhibit performance degradation, indicating limited scalability.

Figure 3: **Data scaling behavior** on the `ogbn-arxiv` and `ogbn-products` with linear probing. $G^2PM$ demonstrates robust improvements with more pre-training data, reflecting superior scalability. In contrast, GraphMAE [22] and BGRL [53] peak at small data ratios and degrade with more data, suggesting overfitting, poor regularization, or inefficient data utilization.

# 1 Introduction

Transformer architectures [59] have emerged as the cornerstone of modern foundation models [7, 13, 2, 57]. Enabled by generative pre-training on massive unlabeled corpora [7, 12], Transformers learn transferable representations that can be efficiently adapted to diverse downstream tasks. This paradigm shift has redefined the landscape of machine learning, powering state-of-the-art systems in natural language processing and computer vision. Prominent examples include large language models (LLMs) [2, 57] and Vision Transformers (ViTs) [13], which showcase the scalability and generalization strength of this approach.

Despite the transformative success of generative Transformer pre-training in text and vision domains, this paradigm has yet to bring comparable breakthroughs in the graph domain, posing a significant barrier to the development of graph foundation models [40, 65, 67]. Most existing approaches to graph pre-training remain grounded in message-passing graph neural networks (MPNNs) [33, 60, 16, 18, 39], which are known to suffer from fundamental limitations: constrained expressive power [77], over-smoothing [49], over-squashing [56], and poor capacity for modeling long-range dependencies [47]. These challenges significantly limit the scalability of MPNNs [43], where increasing the size of the model or training data does not reliably lead to improved performance. As a result, the emergence of scaling laws in graph learning—a critical property in the success of foundation models [28]—remains elusive. Moreover, current graph pre-training techniques are predominantly based on contrastive learning [46, 83, 64, 63, 53, 84, 75, 76, 45, 38], which has been shown to be less capable of learning generalizable semantic representations compared to generative objectives [20, 12, 54, 5, 7]. This reliance further compounds the scalability bottleneck of graph neural networks (GNNs).

In this work, we aim to extend the success of Transformer-based generative pre-training to the graph domain, with the goal of enabling scalable graph representation learning. We consider the effectiveness of Transformers stems from a common principle: *Transformer-based models represent modality-specific instances using sequences of high-level semantic tokens, and apply generative objectives for pre-training*. To realize this paradigm for graphs, we begin by identifying three fundamental challenges that distinguish graph-structured data from Euclidean modalities such as text and images. (1) **Absence of Sequence Structure:** Unlike text or images, which naturally possess ordered or grid-like sequences compatible with Transformer training, graphs lack a sequence (no matter ordered or unordered) for nodes, edges, or subgraphs. This complicates the adaptation of Transformer. (2) **Semantic Granularity:** Graph elements—such as nodes or edges—typically encode low-level semantics, while tokens in language (i.e., words) or vision (i.e., patches) often correspond to higher-level concepts. It is unclear whether Transformers can effectively learn from such low-level representations in a generative fashion. (3) **Scalability Bottlenecks:** Existing Graph Transformers

(GTs) [34, 9, 14] often treat individual nodes as tokens and focus on pairwise relationships, leading to sequence lengths that scale with the number of nodes. Due to the quadratic time complexity of Transformers [29], this design choice limits scalability to large graphs, confining GTs primarily to small graphs such as molecular graphs [47].

These challenges naturally raise a central question: *how can we define sequences of high-level semantic tokens that meaningfully represent a graph instance—whether a node, an edge, or an entire graph?* To answer this, we revisit the core objective of graph learning: understanding key substructures that are predictive of downstream tasks. In many domains, such substructures are inherently semantic. For instance, motifs like triangles in social networks capture stable interpersonal relationships, while benzene rings in molecular graphs encode chemical stability—both serving as informative building blocks for reasoning. Motivated by this intuition, we propose to represent graph instances as sequences of meaningful substructures (Figure 1) and introduce the **G**enerative **G**raph **P**attern **M**achine ($G^2PM$), a Transformer-based generative pre-training framework for graphs. $G^2PM$ tokenizes graphs into sequences of substructures via random walks, and learns representations by reconstructing masked substructures from context. This approach enables the design of pure Transformer models for graphs—free from message-passing—while unlocking scalability through increased model capacity and data volume (Figures 2 and 3). We highlight our contributions:

- We propose $G^2PM$, a generative Transformer pre-training framework that models graph instances as sequences of substructures, entirely eliminating the need for message passing.

- We introduce a random walk-based tokenizer that efficiently extracts semantic substructure patterns, and pair it with a masked substructure prediction task to enable self-supervised learning.

- We conduct a comprehensive design space exploration, offering actionable insights into model architecture and scalability.

- We demonstrate that $G^2PM$ scales effectively: increasing data or model size yields consistent performance gains, echoing the scaling behaviors observed in other domains.

- We validate $G^2PM$ across multiple benchmarks, showing state-of-the-art performance on node/link/graph-level tasks, along with strong generalization in cross-graph transfer tasks.

**Outlook.** While Transformers have redefined learning paradigms in many domains, their influence on graph learning remains nascent. We hope this work sparks further exploration into *non-message-passing* architectures, especially Transformers, as a new foundation for graph representation learning.

## 2 Generative Graph Pattern Machine

Let $\mathcal{G} = (\mathcal{V}, \mathcal{E}, \mathbf{X}, \mathbf{E})$ denote a graph, where $\mathcal{V}$ is the set of nodes with $|\mathcal{V}| = N$, $\mathcal{E} \subseteq \mathcal{V} \times \mathcal{V}$ is the set of edges with $|\mathcal{E}| = E$, and $\mathbf{X}$ and $\mathbf{E}$ represent the node and edge feature matrices, respectively. Each node $v \in \mathcal{V}$ is associated with a feature vector $\mathbf{x}_v \in \mathbb{R}^{d_n}$, and each edge $e \in \mathcal{E}$ is associated with a feature vector $\mathbf{e}_e \in \mathbb{R}^{d_e}$ (when applicable). As illustrated in Figure 4, $G^2PM$ is pre-trained using a masked substructure modeling (MSM) objective in a fully self-supervised fashion.

### 2.1 Graph Representations

Our core idea is to represent any graph instance (node, link, or graph) as a sequence of substructures. In natural language, sequences are constructed using a predefined vocabulary of words or subwords; in vision, raster-scan tokenizers divide images into patch sequences. However, extending this notion of tokenization to graphs poses two key challenges. First, it is non-trivial to define a universal vocabulary of graph substructures, as the semantic relevance of patterns is highly domain-specific. For example, triangle motifs are prominent in social networks, whereas ring structures are more meaningful in molecular graphs. Second, tokenizing graphs by substructure matching incurs high computational cost: subgraph isomorphism is NP-complete [15], rendering it impractical for large-scale graphs—especially when compared to the linear-time tokenization of text and images [51, 13].

**Tokenizer.** To overcome these challenges, we adopt a random walk-based tokenizer that samples substructure patterns on the fly, bypassing the need for a predefined vocabulary. This strategy offers an efficient and scalable tokenization mechanism [68, 81]. As illustrated in Figure 4(b), a random

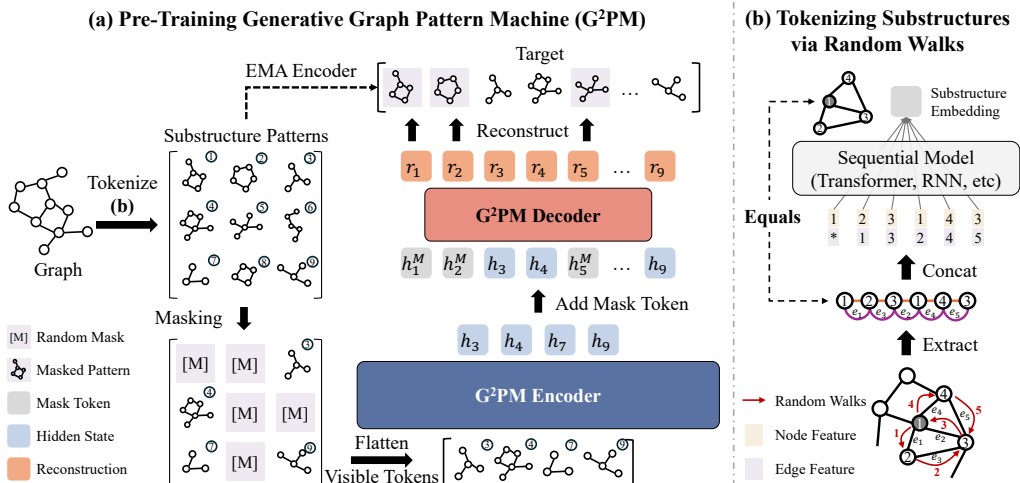

Figure 4: **Overview of G²PM pre-training.** (a) Given a graph instance, we first apply a random walk-based tokenizer to extract substructure patterns. Some patterns are randomly masked and the visible patterns are fed into a G²PM encoder. The encoder outputs are concatenated with special mask tokens and passed to a G²PM decoder to reconstruct the masked substructures. (b) We tokenize substructures by performing random walks over the graph, where each walk represents a substructure, which is proved to be effective [68]. The substructure embeddings are modeled via Transformer.

walk such as $[1, 2, 3, 1, 4, 3]$ corresponds to a diamond-shaped substructure. Formally, we define an unbiased random walk $w$ of fixed length $L$ as a sequence sampled from a Markov chain:

$$P(v_{i+1} \mid v_0, \ldots, v_i) = \frac{\mathbb{1}[(v_i, v_{i+1}) \in \mathcal{E}]}{D(v_i)}, \tag{1}$$

where $D(v_i)$ denotes the degree of node $v_i$. This transition probability enables efficient generation of node sequences from arbitrary starting points.

For each graph instance, we sample $k$ random walks to serve as substructure patterns. To balance locality and global context, we leverage unbiased sampling strategies [44, 68]. Rather than constructing explicit subgraphs from these walks, we treat each as a node sequence and directly encode it using a Transformer, which has been shown to effectively capture structural inductive biases in graphs [68].

Given a node sequence $w = [\mathbf{x}_1, \ldots, \mathbf{x}_m]$, we compute its embedding via a Transformer encoder $f$:

$$\mathbf{p} = f(w) = f([\mathbf{h}_1, \ldots, \mathbf{h}_m]), \quad \mathbf{h}_i = \mathbf{W}\mathbf{x}_i + \mathbf{b}, \tag{2}$$

where $\mathbf{p}$ is the substructure embedding, and $\mathbf{x}_i = [\mathbf{x}_i \| \mathbf{e}_i]$ is the concatenation of node feature $\mathbf{x}_i$ and (optionally) edge feature $\mathbf{e}_i$.

**Eliminating Message-Passing Bottlenecks.** Our tokenizer is entirely message-passing-free, allowing G²PM to bypass key limitations of traditional GNNs. Notably, the random walk-based representations in G²PM effectively capture long-range dependencies [68, 27], exceed the expressiveness of 1-WL tests [55, 69, 42], and mitigate over-smoothing and over-squashing effects [61].

**No Positional Embeddings.** Unlike standard Transformers, which depend on positional encodings, we find that adding position information to node sequences offers no consistent benefit (Table 1b) and incurs cubic time complexity [34]. We thus omit positional embeddings entirely, simplifying the architecture without compromising performance.

**Task-Agnostic Tokenization.** Our tokenizer is applicable across graph-based tasks. For *node-level tasks*, random walks originate from the target node; for *edge-level tasks*, from the endpoints of target edge; and for *graph-level tasks*, from randomly sampled nodes in the graph. This task-agnostic design enables G²PM to adapt seamlessly without requiring specialized modifications.

## 2.2 G²PM Backbone

We adopt the standard Transformer architecture [59] as the backbone for both G²PM encoder and decoder. The input to the Transformer is a sequence of graph substructure tokens $\mathbf{P} = [\mathbf{p}_1, \mathbf{p}_2, \ldots, \mathbf{p}_n]$, where each $\mathbf{p}_i$ denotes the embedding of a sampled substructure. A single Transformer layer operates via a combination of self-attention and feed-forward networks, formally defined as:

$$\mathbf{P}' = \text{FFN}\left(\mathbf{P} + \text{Attn}(\mathbf{P})\right), \quad \text{Attn}(\mathbf{P}) = \text{softmax}\left(\frac{\mathbf{Q}\mathbf{K}^\top}{\sqrt{d_{\text{out}}}}\right)\mathbf{V} \in \mathbb{R}^{n \times d_{\text{out}}}, \tag{3}$$

$$\mathbf{Q} = \mathbf{P}\mathbf{W_Q}, \quad \mathbf{K} = \mathbf{P}\mathbf{W_K}, \quad \mathbf{V} = \mathbf{P}\mathbf{W_V}, \tag{4}$$

where $\mathbf{W_Q}, \mathbf{W_K}, \mathbf{W_V}$ are learnable projection matrices and $d_{\text{out}}$ denotes the dimensionality of the output embeddings. The FFN is implemented as a two-layer multilayer perceptron with non-linearity. Following standard practice, we employ multi-head self-attention [59], where multiple attention mechanisms are applied in parallel and their outputs concatenated. We stack multiple Transformer layers to enhance model capacity. We denote the output of the final ($L$-th) Transformer layer as $\mathbf{P}^L = [\mathbf{p}_1^L, \mathbf{p}_2^L, \ldots, \mathbf{p}_n^L]$, representing the learned contextualized embeddings of graph substructures.

## 2.3 Pre-Training G²PM: Masked Substructure Modeling

To pre-train the G²PM model, we introduce a generative Transformer objective *masked substructure modeling* (MSM). Given a sequence of substructures extracted from a graph, we randomly mask a subset and train the model to reconstruct the masked substructures conditioned on the visible ones. This process follows the conditional factorization principle used in masked language modeling [12]:

$$p(x_1, x_2, \ldots, x_n) = p(x_{/M}) \prod_{t \in M} p(x_t \mid x_{/M}), \tag{5}$$

where $x_{/M}$ denotes the set of visible tokens and $M$ indicates the masked positions.

The key intuition is that meaningful graph substructures exhibit strong interdependencies—such as hierarchical relationships, functional reinforcement, or functional exclusion (Appendix A)—allowing certain substructures to be predictive of others. Concretely, given a tokenized substructure sequence $\mathbf{P} = [\mathbf{p}_1, \ldots, \mathbf{p}_n]$, we randomly retain $k$ visible tokens to form $\mathbf{P}_{\text{vis}}$, while masking the remaining $n - k$ tokens. The visible sequence $\mathbf{P}_{\text{vis}}$ is passed through the G²PM encoder to produce contextual embeddings $\mathbf{H}_{\text{vis}}$. We then insert learnable mask tokens at the masked positions to recover the full-length sequence, and feed the sequence into the G²PM decoder for reconstruction $\mathbf{R} = [\mathbf{r}_1, \ldots, \mathbf{r}_n]$:

$$\mathbf{R} = \text{Decoder}(\mathbf{H}), \quad \mathbf{H} = \text{Add}_{[\text{M}]}(\mathbf{H}_{vis}), \quad \mathbf{H}_{vis} = \text{Encoder}(\mathbf{P}_{vis}). \tag{6}$$

To define the reconstruction targets, we leverage the high-level semantics of substructures. Instead of reconstructing low-level node features or adjacency matrices, we employ an *online encoder* $f_{\text{EMA}}$, an exponential moving average (EMA) version of the substructure encoder, to generate target embeddings $\hat{\mathbf{p}}_i = f_{\text{EMA}}(w_i)$ for each substructure walk $w_i$. This yields the training objective:

$$\mathcal{L} = \frac{1}{n} \sum_{i=1}^{n} \text{is\_masked}(\mathbf{p}_i) \cdot \|\mathbf{r}_i - \text{sg}[\hat{\mathbf{p}}_i]\|_2^2, \quad \hat{\mathbf{p}}_i = f_{\text{EMA}}(w_i), \tag{7}$$

where $\text{is\_masked}(\cdot)$ is an indicator function for masked positions, and $\text{sg}[\cdot]$ denotes the stop-gradient operator, ensuring that targets are fixed during training.

**Adapting to Downstream Task.** For downstream tasks, we discard the decoder and append a linear prediction head to the G²PM encoder. Each graph instance is tokenized into a sequence of substructure embeddings $\mathbf{P} = [\mathbf{p}_1, \mathbf{p}_2, \ldots, \mathbf{p}_n]$, which is processed by the $L$-layer encoder to produce contextual representations $\mathbf{P}^L = [\mathbf{p}_1^L, \ldots, \mathbf{p}_n^L]$. We then apply mean pooling followed by the task head for final prediction $\hat{\mathbf{y}} = \text{Head}\left(\frac{1}{n} \sum_{i=1}^{n} \mathbf{p}_i^L\right)$.

**Augmenting Substructure Patterns.** To improve robustness and representation quality, we adopt a mixed augmentation strategy inspired by corruption-agnostic pre-training [20, 71]. We define four augmentations in two categories: *Feature-level*—(1) feature masking (zeroing a subset of features) and (2) node masking (masking entire nodes); and *Structure-level*—(3) substructure corruption (dropping nodes within a walk) and (4) substructure injection (replacing walk nodes with random nodes). During training, we apply one feature-level and one structure-level augmentation per instance to promote generalization across diverse perturbations.

| Dim | # Params | Arxiv | HIV |
|---|---|---|---|
| 64 | ∼0.4M | 68.2 | 69.7 |
| 128 | ∼1.5M | 70.5 | 70.9 |
| 256 | ∼6.2M | 71.2 | 74.1 |
| 512 | ∼26.6M | 72.1 | 76.4 |
| 768 | ∼64.5M | **72.3** | **78.7** |

(a) **Model dimension.** G$^2$PM exhibits strong scalability with increased hidden dimension sizes.

| Encoder | Arxiv | HIV |
|---|---|---|
| Transformer | 72.3 | **78.7** |
| GRU | 63.4 | 72.3 |
| GIN | 71.0 | 73.2 |
| MEAN | 70.3 | 70.4 |
| + Node PE | **72.4** | 78.6 |

(b) **Tokenization.** Transformer is the most expressive encoder for modeling substructures.

| [Mask] | Arxiv | HIV |
|---|---|---|
| Learnable | **72.3** | **78.7** |
| Fixed | 71.8 | 76.9 |
| Random | 67.6 | 71.6 |
| Sampling | 67.1 | 74.0 |

(c) **Mask token.** Using a learnable mask token leads to higher accuracy compared to other tokens.

| Aug. | Arxiv | HIV |
|---|---|---|
| Mixed | **72.3** | **78.7** |
| Feature mask $p = 0.2$ | 69.8 | 73.2 |
| Node mask $p = 0.2$ | 70.9 | 75.5 |
| Substructure corrupt $p = 0.8$ | 71.3 | 74.4 |
| Substructure inject $p = 0.8$ | 72.1 | 72.4 |
| None | 70.8 | 73.9 |

(d) **Data augmentation.** Mixed augmentation significantly improves downstream performance.

| Case | Arxiv | HIV |
|---|---|---|
| EMA Emb. | 72.3 | 78.7 |
| Feat. (mean) | 71.4 | 73.5 |
| Feat. (concat) | 71.2 | 64.0 |
| L2 to L1 loss | 72.1 | 72.3 |
| + Topo. Recon. | **72.9** | **78.9** |

(e) **Reconstruction target.** High-level target is more effective than feature-level reconstruction.

| $\alpha$ | Update Every | Arxiv | HIV |
|---|---|---|---|
| 0.9 | 10 | 70.8 | 74.3 |
| 0.99 | 10 | 72.3 | 78.7 |
| 0.999 | 10 | 72.1 | 76.6 |
| 0.99 | 5 | 72.2 | 78.1 |
| 0.99 | 20 | 72.3 | 76.6 |
| w/o EMA Update | | 25.2 | 69.1 |

(f) **Online encoder.** Maintaining an EMA-updated online encoder is crucial in generating targets.

Table 1: **G$^2$PM ablation experiments** on ogbn-arxiv and ogbg-HIV. We report linear probe accuracy (%) and use gray to indicate default settings. The detailed hyper-parameters are in Appendix B.

# 3 Design Spaces and Insights

We conduct a comprehensive ablation study to provide more insights about the model design.

## 3.1 Model Architecture

**Model Dimension.** We set the default hidden dimension to 768, with a 3-layer encoder and a 1-layer decoder. As shown in Table 1a, increasing model size consistently improves performance, mirroring scaling trends observed in language and vision domains [7, 13]. Unlike message-passing models that often plateau with scale, G$^2$PM continues to benefit from increased parameterization.

| Arch. | Enc. | Dec. | # Enc. Params | # Dec. Params | Arxiv | HIV |
|---|---|---|---|---|---|---|
| MAE | 1 | 1 | ∼7.1M | ∼7.1M | 71.1 | 75.9 |
| MAE | 2 | 1 | ∼14.2M | ∼7.1M | 72.2 | 77.6 |
| MAE | 3 | 1 | ∼21.3M | ∼7.1M | **72.3** | **78.7** |
| MAE | 3 | 2 | ∼21.3M | ∼14.2M | 72.0 | 71.9 |
| SimMIM | 3 | 1 | ∼21.3M | ∼0.6M | 71.1 | 74.7 |
| SimMIM | 3 | 2 | ∼21.3M | ∼1.2M | 71.7 | 74.1 |

(a) **Model architecture.** Our G$^2$PM design (MAE-style [20]) outperforms SimMIM-style [71] pre-training.

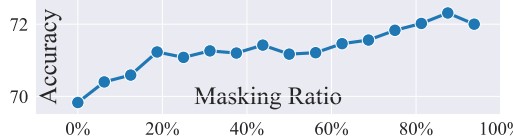

(b) **Masking ratio.** A large masking ratio leads to better performance with more challenging tasks.

Figure 5: **More G$^2$PM ablation experiments**.

**Model Layer.** Table 5a shows that deeper encoders enhance performance, while increasing decoder capacity provides limited or negative returns, particularly under the MAE-style masked modeling framework. We attribute this to the simplicity of substructure reconstruction, where expressive decoders can lead to overfitting. Replacing the MAE-style design [20] with a SimMIM-style variant [71] further degrades performance. This suggests that MAE-style sparsity encourages the encoder to learn stronger context-aware representations, which better align with the inductive bias of graph data.

## 3.2 Reconstruction Target

**Target.** By default, G$^2$PM reconstructs the semantic embedding of each masked substructure generated by an EMA-updated encoder. As shown in Table 1e, this choice consistently yields the strongest performance, supporting our hypothesis that substructures encode transferable semantic information. We compare this approach to two low-level alternatives. Specifically, we represent each substructure as a node sequence $w = [x_1, \ldots, x_n]$, where each $x_i = [\mathbf{x}_i \| \mathbf{e}_i]$ is the concatenation of node and optional edge features. We experiment with (1) mean-pooling the node features over the walk, and (2) concatenating them directly. Both approaches underperform the semantic embedding target, suggesting that low-level supervision lacks the abstraction necessary for graph generalization.

**Loss.** We also evaluate the impact of loss functions. Replacing $\ell_2$ loss with $\ell_1$ leads to a notable drop in graph classification accuracy, suggesting that outlier-sensitive objectives better capture nuanced substructure semantics. Additionally, we incorporate anonymous walks [24] following Wang et al. [68], using a parallel head to reconstruct topological patterns. This auxiliary objective consistently boosts performance, reinforcing the utility of substructures as fundamental modeling units.

**Online Encoder.** We ablate the momentum $\alpha$ and update the frequency of the EMA encoder (Table 1f). Our default setting ($\alpha = 0.99$, update every 10 steps) performs best. Lower momentum values degrade performance, with $\alpha = 0$ (i.e., no EMA) leading to divergence and collapse, highlighting the importance of stable, slowly evolving targets during training.

### 3.3 Tokenization

**Substructure Encoder.** To embed substructure sequences (e.g., random walks), we adopt a Transformer encoder, which models global dependencies among all nodes in the sequence. As shown in Table 1b, Transformer outperforms alternatives such as mean pooling, GRU [11], and GIN [73]. We attribute this to several factors: (1) mean pooling is a restricted, less expressive form of attention in Transformer; (2) GRUs suffer from optimization instability due to vanishing or exploding gradients [85]; and (3) using GIN requires converting walks into subgraphs, reintroducing message-passing and its known limitations.

**Positional Embedding.** We also explore adding positional embeddings (PEs) to node features—using Laplacian PE on ogbn-arxiv and random walk PE on ogbg-HIV [47]. However, we observe no consistent performance gain. This may be due to the tokenization scheme: since each token represents an entire substructure, global node positions are less relevant. Moreover, computing PEs incurs cubic time complexity in the node number [34], making them impractical for large-scale graphs.

### 3.4 Augmentation

Table 1d reports the impact of various augmentation strategies. In general, augmentation improves performance, with the mixed strategy yielding the best results, highlighting the value of combining perturbations from different perspectives. We further analyze individual augmentation types. Among feature-level augmentations, node masking outperforms feature masking, likely because reconstructing full-node semantics from context is more challenging—and thus more informative—than recovering masked dimensions. For structure-level perturbations, substructure corruption performs better on graph-level tasks (e.g., molecular graphs), while substructure injection is more effective on node-level tasks (e.g., academic networks). This distinction stems from how each augmentation impacts semantics: corruption distorts but retains the original context, whereas injection alters substructure identity, which may hurt graph-level tasks where substructure semantics are functionally meaningful. Notably, using only a single augmentation type yields limited improvements, reinforcing the importance of diversity in corruption strategies for generalizable representation learning.

### 3.5 Masking

**Masking Ratio.** Figure 5b shows that a high masking ratio consistently improves performance, echoing trends observed in vision [20]. This is likely because random walks often produce redundant or noisy substructures. High masking reduces such redundancy, resulting in a more challenging and informative learning signal that encourages the model to capture high-level semantics.

**Masking Token.** The choice of masking token significantly affects performance (Table 1c). By default, we use a single learnable mask token, which outperforms alternatives. Fixed tokens degrade performance due to their inflexibility, while random tokens or sampled embeddings from other substructures introduce noise that confuses the model. These findings suggest that a consistent, learnable mask token provides the strongest and cleanest supervision signal during reconstruction.

## 4 Comparisons to State-of-The-Art Models

**Node Classification on homophily graphs.** We evaluate $G^2PM$ on a suite of homophily graphs of varying scales, including Pubmed, Photo, Computers, WikiCS, Flickr, ogbn-arxiv, and ogbn-products (see Table 2 for dataset statistics). Pre-training is conducted on the full graph. We adopt a linear probe setup: node embeddings are frozen after pre-training and used to train a separate classifier, where we take 10/10/80 random split for Pubmed, Photo, and Computers, and the official split for the remaining datasets. Accuracy is used as the evaluation metric. We compare $G^2PM$ to supervised GAT [60], non-message-passing GPM [68], contrastive methods (GCA [84], BGRL [53], CCA-SSG [78]), and generative methods (GraphMAE [22], GraphMAE 2 [23], S2GAE [52], Bandana [82]).

Table 2: **Node classification** results on homophily graphs. **Boldface** and underline indicate the best and sub-best self-supervised methods, and A.R. is the average ranking.

| | | Pubmed | Photo | Computers | WikiCS | Flickr | Arxiv | Products | A.R. |
|---|---|---|---|---|---|---|---|---|---|
| | # Nodes | 19,717 | 7,650 | 13,752 | 11,701 | 89,250 | 169,343 | 2,449,029 | - |
| | # Edges | 88,648 | 238,162 | 491,722 | 431,206 | 899,756 | 2,315,598 | 123,718,024 | - |
| **Supervised** | GAT [60] | 83.1 ± 0.3 | 91.9 ± 0.5 | 87.9 ± 0.5 | 76.9 ± 0.8 | 50.7 ± 0.3 | 72.10 ± 0.13 | 79.45 ± 0.59 | 5.7 |
| | GPM [68] | 84.7 ± 0.1 | 92.7 ± 0.3 | 90.0 ± 0.4 | 80.2 ± 0.4 | 52.2 ± 0.2 | 72.89 ± 0.68 | 82.62 ± 0.39 | 1.3 |
| **Contrastive** | GCA [84] | 83.3 ± 0.5 | 92.4 ± 0.2 | 87.1 ± 0.2 | 77.4 ± 0.1 | 49.0 ± 0.1 | 71.23 ± 0.09 | 78.39 ± 0.03 | 6.9 |
| | BGRL [53] | 83.9 ± 0.3 | 92.5 ± 0.2 | 88.2 ± 0.2 | 77.5 ± 0.8 | 49.7 ± 0.2 | 70.51 ± 0.03 | 78.59 ± 0.02 | 5.7 |
| | CCA-SSG [78] | 81.8 ± 0.5 | 91.8 ± 0.6 | 88.6 ± 0.3 | 75.3 ± 0.8 | 47.5 ± 0.2 | 71.24 ± 0.20 | 75.27 ± 0.05 | 8.6 |
| **Generative** | GraphMAE [22] | 81.0 ± 0.5 | 92.0 ± 0.3 | **89.2 ± 0.5** | 77.1 ± 0.5 | 50.5 ± 0.1 | 71.75 ± 0.17 | 78.89 ± 0.01 | 6.0 |
| | GraphMAE 2 [23] | 81.3 ± 0.4 | 92.4 ± 0.2 | 88.3 ± 0.9 | 77.6 ± 0.4 | 50.4 ± 0.1 | 71.89 ± 0.03 | 79.33 ± 0.01 | 5.4 |
| | S2GAE [52] | 80.1 ± 0.5 | 91.4 ± 0.1 | 85.3 ± 0.1 | 75.3 ± 0.8 | 48.1 ± 0.8 | 67.77 ± 0.36 | 76.70 ± 0.03 | 10.3 |
| | Bandana [82] | 83.5 ± 0.5 | 91.4 ± 0.7 | 87.7 ± 0.2 | 77.3 ± 0.3 | 47.9 ± 0.6 | 71.09 ± 0.24 | 77.68 ± 0.05 | 8.1 |
| | G²PM w/o Pretrain | 83.9 ± 0.2 | 92.8 ± 0.2 | 87.1 ± 0.3 | 78.5 ± 0.4 | 50.7 ± 0.1 | 69.64 ± 0.08 | 76.90 ± 0.16 | 6.0 |
| | G²PM | **84.3 ± 0.1** | **92.9 ± 0.2** | 88.8 ± 0.3 | **79.0 ± 0.4** | **51.0 ± 0.0** | **72.31 ± 0.07** | **80.56 ± 0.01** | 2.0 |

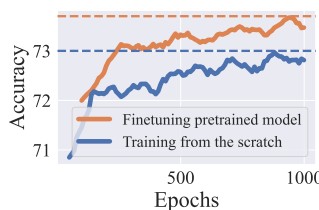

Figure 6: **Convergence curves** under fully finetuning on Arxiv dataset. We compare training from scratch versus fine-tuning the pre-trained model.

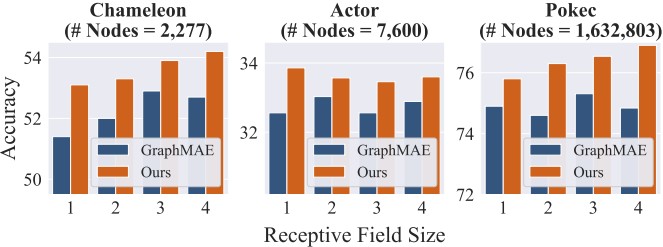

Figure 7: **Node classification** results on heterophily graphs under varying receptive field sizes. For GraphMAE, the receptive field is controlled by the number of model layers; in our method, it is controlled by the random walk length, where a model layer of size $k$ corresponds to a walk length of $2^k$.

As shown in Table 2, G²PM outperforms all self-supervised baselines and ranks second overall—only behind the supervised GPM—achieving an average rank of 2.0. Notably, on the large-scale ogbn-products dataset, G²PM achieves a substantial performance gain (80.56 vs. 79.33), highlighting its scalability. Ablation shows that on small graphs ($\leq$ 100K nodes), the model performs well even without pre-training, likely due to its sufficient capacity. However, on larger datasets like ogbn-arxiv and ogbn-products, pre-training proves crucial for capturing fine-grained structural patterns.

**Node Classification on Heterophily Graphs.** We evaluate G²PM on heterophily graphs: Chameleon, Actor, and Pokec (over 1M nodes). Results are shown in Figure 7. We compare against GraphMAE, varying the receptive field via the number of GNN layers (for GraphMAE) and random walk length (for G²PM ). G²PM consistently outperforms GraphMAE across datasets and benefits from increased receptive field size, demonstrating its ability to capture non-local information critical for heterophilous settings. On Actor, however, performance shows no clear correlation with receptive field, suggesting that key signals in this collaboration network are primarily localized.

**Link Prediction.** To further demonstrate the effectiveness of our pretraining approach, we follow Wang et al. [68] to show the link prediction results. We use three datasets, including Cora, Pubmed, and ogbl-collab, and follow a standard 80/5/15 split for training, validation, and test sets. For evaluation, we report Hit@20 on Cora and Pubmed, and Hit@50 on ogbl-collab. The results are summarized in Table 3, where we compare our method G²PM with a basic super-

Table 3: **Link prediction results** on three widely-used benchmarks.

| | Cora | Pubmed | ogbl-Collab |
|---|---|---|---|
| # Nodes | 2,708 | 19,717 | 235,868 |
| # Edges | 10,556 | 88,648 | 2,570,930 |
| Metric | Hit@20 | Hit@20 | Hit@50 |
| GCN [33] | 84.1 ± 1.2 | 85.1 ± 3.8 | 44.8 ± 1.1 |
| GraphMAE [22] | 87.6 ± 1.4 | 87.1 ± 3.1 | 46.5 ± 0.9 |
| G²PM | **90.4 ± 0.8** | **88.0 ± 3.8** | **47.1 ± 0.6** |

vised GCN [33] and a widely used graph pretraining method, GraphMAE [22]. As shown, GraphMAE significantly outperforms GCN, benefiting from its ability to utilize unlabeled data during pretraining. Notably, G²PM achieves the best performance across all datasets, likely due to its capacity to model substructures that capture latent connectivity patterns between nodes.

Table 4: **Graph classification** results on molecular and social networks. **Boldface** and underline indicate the best and sub-best self-supervised methods, and A.R. is the average ranking.

| | | HIV | PCBA | Sider | MUV | ClinTox | IMDB-B | REDDIT-M12K | A.R. |
|---|---|---|---|---|---|---|---|---|---|
| | # Graphs | 41,127 | 437,929 | 1,427 | 93,087 | 1,478 | 1,000 | 11,929 | - |
| | # Nodes | ∼25.5 | ∼26.0 | ∼33.6 | ∼24.2 | ∼26.1 | ∼19.8 | ∼391.4 | - |
| | # Edges | ∼27.5 | ∼28.1 | ∼70.7 | ∼52.6 | ∼55.5 | ∼193.1 | ∼913.8 | - |
| Supervised | GIN [73] | 75.8 ± 0.8 | 70.3 ± 0.3 | 57.7 ± 0.8 | 74.4 ± 0.9 | 83.4 ± 0.6 | 73.3 ± 0.5 | 39.4 ± 1.4 | 6.3 |
| | GPM [68] | 77.0 ± 0.9 | 75.1 ± 0.3 | 59.0 ± 0.0 | 74.6 ± 1.4 | 82.4 ± 0.3 | 82.7 ± 0.5 | 43.1 ± 0.3 | 3.0 |
| Contrastive | GraphCL [75] | 75.5 ± 0.3 | 72.4 ± 2.1 | 57.3 ± 0.9 | 68.3 ± 2.6 | 82.9 ± 0.3 | 71.1 ± 0.4 | 37.9 ± 2.4 | 8.0 |
| | JOAO [76] | 76.8 ± 0.3 | 73.4 ± 1.5 | 58.5 ± 0.5 | 72.3 ± 1.0 | 82.2 ± 0.3 | 70.2 ± 3.1 | 39.9 ± 0.6 | 6.0 |
| | MVGRL [19] | 75.7 ± 0.7 | 70.4 ± 2.1 | 60.5 ± 0.6 | 71.5 ± 1.2 | 83.6 ± 0.2 | 74.2 ± 0.7 | 39.5 ± 1.8 | 5.7 |
| | InfoGCL [72] | 77.3 ± 0.6 | 74.6 ± 0.7 | 58.7 ± 0.7 | 73.4 ± 1.0 | 80.3 ± 0.7 | 75.1 ± 0.9 | 39.3 ± 0.5 | 5.4 |
| Generative | GraphMAE [22] | 77.8 ± 0.9 | 73.2 ± 1.4 | 60.6 ± 0.0 | 73.7 ± 0.8 | 84.8 ± 0.5 | 75.5 ± 0.7 | 37.6 ± 2.5 | 4.1 |
| | S2GAE [52] | 75.6 ± 0.8 | 72.9 ± 0.0 | 58.0 ± 0.9 | 71.6 ± 0.8 | 80.6 ± 0.4 | 75.8 ± 0.6 | 37.9 ± 1.8 | 7.0 |
| | $G^2PM$ w/o Pretrain | 69.8 ± 0.1 | 68.4 ± 0.0 | 58.8 ± 0.3 | 66.3 ± 1.4 | 80.0 ± 1.8 | 80.0 ± 0.8 | 37.5 ± 0.3 | 8.3 |
| | $G^2PM$ | **78.7 ± 0.1** | **75.6 ± 0.1** | **61.2 ± 0.2** | **75.7 ± 0.4** | **86.6 ± 0.8** | **83.0 ± 0.8** | **41.8 ± 0.3** | 1.0 |

Table 5: **Cross-domain transferability** performance across diverse source and target datasets. Parentheses indicate the performance gap compared to training from scratch on the target graph.

| Source | Arxiv | | HIV | |
|---|---|---|---|---|
| Target | Products | HIV | Arxiv | PCBA |
| GNN [60, 73] | 78.3 (1.2 ↓) | 70.1 (5.7 ↓) | 71.1 (1.0 ↓) | 71.9 (1.6 ↑) |
| GPM [68] | 82.0 (0.6 ↓) | 74.3 (2.7 ↓) | 71.4 (1.5 ↓) | 76.4 (1.3 ↑) |
| BGRL [53] | 78.8 (0.2 ↑) | 72.5 (3.8 ↓) | 68.6 (1.9 ↓) | 72.9 (0.6 ↓) |
| GraphMAE [22] | 77.5 (1.4 ↓) | 74.7 (3.1 ↓) | 69.9 (1.9 ↓) | 73.4 (0.2 ↑) |
| $G^2PM$ | **81.3 (0.7 ↑)** | **76.8 (1.9 ↓)** | **72.6 (0.3 ↑)** | **77.9 (2.3 ↑)** |

Table 6: **Cross-domain pre-training** results on text-attributed graphs processed by [37], where node features are aligned via a textual encoder.

| Pretrain | Arxiv + FB15K237 + ChemBL | | |
|---|---|---|---|
| Downstream | Arxiv (Academia) | FB15K237 (Knowledge Graph) | HIV (Molecule) |
| BGRL [53] | 70.8 ± 0.2 | 86.5 ± 0.3 | 68.5 ± 1.6 |
| GraphMAE [22] | 70.3 ± 0.3 | 87.8 ± 0.4 | 64.1 ± 0.5 |
| OFA [37] | 71.4 ± 0.3 | 84.7 ± 1.3 | 72.0 ± 1.6 |
| GFT [65] | 71.9 ± 0.1 | **89.3 ± 0.2** | 72.3 ± 2.0 |
| $G^2PM$ | **72.5 ± 0.1** | 88.9 ± 0.5 | **74.1 ± 1.3** |

**Convergence Curves.** Figure 6 compares the training dynamics of models trained from scratch versus those finetuned from a pre-trained $G^2PM$ checkpoint on ogbn-arxiv. Pre-training leads to consistently better performance, indicating that structural knowledge acquired during pre-training accelerates convergence and improves generalization. As expected, accuracy improves with additional training epochs in both settings.

**Graph Classification.** We evaluate $G^2PM$ on seven datasets: five molecular graphs (HIV, PCBA, SIDER, MUV, ClinTox) and two social networks (IMDB-B, Reddit-M12K). Dataset statistics and results are summarized in Table 4. We use public splits for molecular graphs and 80/10/10 random splits for social networks, following a linear probe protocol. Baselines include supervised GAT [60], non-message-passing GPM [68], contrastive methods (GraphCL [75], JOAO [76], MVGRL [19], InfoGCL [72]), and generative models (GraphMAE [22], S2GAE [52]). $G^2PM$ consistently achieves the best performance across all datasets. In contrast, the model without pre-training performs worst on average, highlighting the importance of capturing domain-specific substructure distributions to support discriminative generalization.

## 5 Cross-Domain Graph Learning

**Cross-Domain Transfer.** We evaluate the cross-domain transferability of $G^2PM$ in Table 5, measuring generalization under distribution shifts across domains and tasks. In this setting, the model is pre-trained on a source graph and fully fine-tuned on a target graph. Since feature spaces differ across graphs, we introduce a learnable linear projection layer before the pre-trained encoder, which is jointly finetuned during transfer. Baselines include GNNs (GAT [60] for ogbn-arxiv and GIN [73] for HIV), as well as GPM [68], BGRL [53], and GraphMAE [22]. $G^2PM$ achieves the best transfer results, showing positive gains in 3 out of 4 setups. In contrast, message-passing methods only transfer well across closely related domains (e.g., HIV → PCBA). We attribute this to the ability to learn transferable structural patterns, enabling generalization even across domain and task boundaries (e.g., Arxiv → HIV), while message-passing remains sensitive to subtle structural shifts [66].

**Cross-Domain Pre-Training.** Table 6 presents results on cross-domain pre-training using three diverse graph types: Arxiv (academic network), FB15K237 (knowledge graph), and HIV (molecular graph) [37]. All graphs are text-attributed; we use a shared textual encoder to project node descriptions

into a unified embedding space, enabling a single model to operate across domains. We compare $G^2PM$ to pre-training baselines (BGRL [53], GraphMAE [22]) and graph foundation models (OFA [37], GFT [65]). $G^2PM$ achieves the best or second-best performance across all datasets, highlighting its superior ability to learn transferable substructure representations, even under significant domain shifts—where message-passing-based methods often struggle.

## 6    Related Works

**Graph Transformers.** Inspired by the success of Transformers in vision and language [12, 7, 57, 20, 13], several works have extended this architecture to graphs [34, 14, 47, 21, 9, 8, 79, 80]. Typically, graph Transformers treat nodes as tokens and apply self-attention over all pairs. However, the quadratic complexity of node attention limits their scalability, making them impractical for large graphs [70]. To address this, recent works propose tokenizing graphs via substructures such as random walks or motifs [74, 24], enabling sequence-based modeling [68, 9, 21, 30]. For example, GPM [68] applies a ViT-style architecture over substructure sequences. Building on this idea, our work explores generative pretraining using a Transformer over substructures, entirely removing message passing.

**Generative Pretraining.** Generative pretraining has driven major advances in vision and language. These methods predict masked content from visible context, enabling learning from large-scale unlabeled data and powering modern foundation models [57]. This paradigm has extended to other domains (e.g., video, audio, biology), but its impact on graphs remains limited. More recent works apply generative approaches to graphs (e.g., VGAE [32], GraphMAE [22], GraphMAE2 [23], MaskGAE [35], G2PT [10]), but focus on reconstructing low-level signals such as nodes or links, and typically rely on message-passing GNNs, limiting their expressiveness and scalability. In contrast, $G^2PM$ introduces a fully Transformer-based, message-passing-free framework that models high-level semantic substructures, unlocking better scalability and generalization.

**Random Walks on Graphs.** Random walks have been widely used to represent substructures, from early unsupervised embeddings [44, 17, 74, 24] to recent deep models [68, 81, 27, 41, 55, 26, 25, 31]. These methods have been shown to break key limitations of message passing, improving expressiveness [69, 42], modeling long-range dependencies [41, 55], and mitigating over-smoothing and over-squashing [62]. However, these random walk-based methods often focus on global patterns while overlooking localized structures [55]. $G^2PM$ uses random walks to sample substructures and design the MSM task to automatically balance the contributions between localized and globalized substructures.

## 7    Conclusion

We introduce $G^2PM$, a Transformer-based generative pretraining framework for graphs that operates entirely without message passing. Unlike traditional GNNs, $G^2PM$ scales effectively with both data and model size, and consistently improves performance across tasks. Through extensive ablations and experiments, we demonstrate its expressiveness, transferability, and potential as a scalable backbone for graph representation learning. While $G^2PM$ leverages unordered substructure sequences—suitable for masked-token prediction—extending it to ordered sequences may enable next-token prediction, further improving scalability. Additionally, our use of random walks as an online tokenizer opens up future directions in designing learnable and adaptive substructure tokenizers for graphs.

## Acknowledgement

The work was partially supported by the NSF under grants IIS-2533550, IIS-2321504, IIS-2340346, IIS-2217239, IIS-2528540, CNS-2426514, and CMMI-2146076, ND-IBM Tech Ethics Lab Program, Notre Dame Strategic Framework Research Grant (2025), and Notre Dame Poverty Research Package (2025). Any expressed opinions, findings, and conclusions or recommendations are those of the authors and do not necessarily reflect the views of the sponsors.

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

# A  From the Perspective of Substructure Dependencies

To understand how G$^2$PM work, we take a functional correlations among substructures to inform both model design and training strategies.

**Hierarchical Dependencies.** Substructures often compose one another hierarchically. For example, a triangle (3-cycle) may serve as a fundamental building block for larger cliques, while a square (4-cycle) can be part of more complex motifs such as diamonds. This compositionality implies that the presence of smaller motifs potentially offer predictive signals for larger structures—and vice versa. From a modeling perspective, this hierarchical property allows the model to infer partially masked substructures based on their visible tokens.

**Functional Reinforcement.** Certain substructures statistically co-occur due to the generative processes underlying different graph types [48]. For instance, in social and citation networks, a high density of triangles often correlates with the existence of higher-order cliques [6, 4]. Similarly, the frequent appearance of short chains may indicate the potential for longer cycle [36, 50]. We refer to this pattern as *functional reinforcement*, where the presence of one motif increases the likelihood of encountering another. This mutual reinforcement reflects domain structural priors and can be leveraged during pre-training to build a predictive inductive bias across motifs.

**Functional Exclusion.** Conversely, some substructures exhibit negative correlations, reflecting competition for structural space within the graph. We term this phenomenon *functional exclusion*. For example, a node embedded within multiple triangle-like motifs is likely part of a densely connected community, making it less probable to support hub-like star patterns [3, 58]. Likewise, graphs with tree-like branches typically lack the density required to support large cliques [1]. Understanding such exclusion enables the model to refine its prediction space: it can down-weight structurally incompatible or redundant patterns during reconstruction.

These inter-substructure dependencies serve dual purposes. On one hand, they enrich the predictive landscape by offering complementary structural cues; on the other hand, they enable the model to disambiguate noisy or redundant substructure tokens by reasoning over motif co-occurrence patterns. By explicitly modeling these dependencies, we endow the system with the capacity to generalize from partial observations and regularize against overfitting to spurious or dataset-specific artifacts.

# B  Implementation Details

## B.1  Environments

Most experiments are conducted on Linux servers equipped with four Nvidia A40 GPUs. The models are implemented using PyTorch 2.4.0, PyTorch Geometric 2.6.1, and PyTorch Cluster 1.6.3, with CUDA 12.1 and Python 3.9.

## B.2  Training Details

Table 7: Default hyper-parameter settings.

| Hyper-parameter | Value | Hyper-parameter | Value |
|---|---|---|---|
| Batch Size | 256 | Gradient Clip | 1 |
| Hidden Dimension | 768 | Optimizer Beta 1 | 0.9 |
| Number of Heads | 12 | Optimizer Beta 2 | 0.005 |
| Number of Encoder Layers | 3 | Min LR | 1e-07 |
| Number of Decoder Layers | 1 | Warmup LR | 1e-07 |
| EMA Momumtum | 0.99 | Scheduler | Cosine |
| EMA Update Every | 10 | Warmup Epochs | 1 |
| Dropout | 0.3 | Linear Probe LR | 0.01 |
| Weight Decay | 0.05 | Linear Probe Weight Decay | 0.001 |

In our setup, we use the AdamW optimizer with weight decay and set the number of epochs as 100. All experiments are conducted five times with different random seeds. The batch size is set to 256 by default. We present detailed default setup in Table 7.

## B.3 Model Configurations

We perform hyperparameter tuning over the following ranges: learning rate $\{1e-3, 7e-4, 5e-4, 3e-4, 1e-4, 7e-5, 5e-5, 3e-5, 1e-5\}$, pattern size $\{4, 8, 16\}$, feature augmentation ratio $p_{\text{feat}} \in [0.0, 0.9]$, and substructure augmentation ratio $p_{\text{sub}} \in [0.0, 0.9]$. The final selected hyper-parameters are reported in Table 8.

Table 8: Dataset-specific hyper-parameter settings.

|  | Pubmed | Photo | Computers | WikiCS | Flickr | Arxiv | Products |
|---|---|---|---|---|---|---|---|
| LR | 3e-5 | 1e-5 | 7e-5 | 1e-5 | 5e-5 | 3e-4 | 3e-4 |
| Feature Aug. $p_{feat}$ | 0.7 | 0.6 | 0.2 | 0.9 | 0.5 | 0.0 | 0.0 |
| Substructure Aug. $p_{struct}$ | 0.3 | 0.8 | 0.3 | 0.9 | 0.1 | 0.8 | 0.8 |
| Pattern Size | 8 | 8 | 8 | 8 | 6 | 8 | 8 |

|  | HIV | PCBA | Sider | MUV | ClinTox | IMDB-B | REDDIT-M12K |
|---|---|---|---|---|---|---|---|
| LR | 3e-5 | 1e-5 | 1e-4 | 5e-5 | 5e-5 | 5e-4 | 3e-4 |
| Feature Aug. $p_{feat}$ | 0.1 | 0.0 | 0.0 | 0.0 | 0.0 | 0.3 | 0.0 |
| Substructure Aug. $p_{struct}$ | 0.3 | 0.8 | 0.7 | 0.2 | 0.0 | 0.3 | 0.6 |
| Pattern Size | 8 | 16 | 8 | 4 | 4 | 8 | 8 |

