# OpenReview forum: "Generative Graph Pattern Machine"
_NeurIPS.cc/2025/Conference — NeurIPS 2025 poster_

### Official Review · Reviewer_85SU · 2025-06-29

**Clarity:** 3
**Significance:** 2
**Originality:** 2
**Rating:** 4
**Confidence:** 3

**Summary:**

This paper introduces G²PM (Generative Graph Pattern Machine), a Transformer-based framework for graph representation learning that bypasses traditional message-passing GNNs. G²PM tokenizes graphs into sequences of substructures (via random walks) and employs masked substructure modeling (MSM) for generative pre-training. The method demonstrates scalability, outperforming existing approaches on tasks like node classification, graph classification, and cross-domain transfer.

**Questions:**

1- You use a fixed tokenizer based on random walks. Would a learnable tokenizer—such as one trained via reinforcement learning or self-supervision—further improve performance?

2- For billion-edge graphs, what modifications would make G²PM practical (e.g., graph sampling, attention approximations)?

3- The paper relies on random walks for substructure extraction, which have known limitations in capturing certain graph properties. How does G²PM address scenarios where key higher-order patterns (like large cliques or complex cycles) aren't sufficiently captured by walk-based sampling?

4- Could you provide more insight into why positional embeddings show no benefit in your framework?

**Ethical Concerns:**

["NO or VERY MINOR ethics concerns only"]

**Final Justification:**

My questions and concerns were resolved.

**Limitations:**

---

**Paper Formatting Concerns:**

---

**Quality:**

2

**Strengths And Weaknesses:**

Strengths:

1- The study presents a comprehensive experimental evaluation, encompassing:
    - node classification benchmarks, including large-scale datasets like OGBN-ArXiv
    - graph classification tasks spanning molecular and social networks
    - Cross-domain transfer experiments
    - In-depth ablation studies

2- The results demonstrate consistent performance improvements with model scaling, achieving gains up to 60 million parameters—supporting the paper’s core hypothesis.


Weaknesses:

1- Theoretical Foundations: Although the empirical results are compelling, the theoretical underpinnings could be strengthened through:
   - formal examination of the expressive power of substructure sequences
   - theoretical analysis comparing the approach to message-passing methods

2- Computational Efficiency: Several key aspects related to scalability are not thoroughly addressed:
   - The memory consumption when applied to large graphs
   - The computational overhead compared to message-passing baselines
   - How the length of walks affects overall computational cost

3- The Related Work section lacks a clear and explicit discussion of how G²PM fundamentally differs from and advances beyond previous substructure-based approaches in a way that highlights its unique contributions.

---

> ### Author Rebuttal · Authors · 2025-07-30
>
> Dear reviewer,
>
> We sincerely thank the reviewer for the detailed and thoughtful evaluation of our work. We appreciate the recognition of G²PM’s comprehensive experimental validation, its consistent scalability, and its strong performance across node classification, graph classification, and cross-domain transfer tasks. We aim to clarify our contributions, address the scalability and expressiveness of G²PM in more detail, and respond to specific questions around tokenizer design, walk limitations, positional embeddings, and deployment at billion-edge scale.
>
> > Q1: The theoretical underpinnings could be strengthened through: (1) formal examination of the expressive power of substructure sequences; (2) theoretical analysis comparing the approach to message-passing methods
> >
>
> Thank you for highlighting this important point. Our primary goal in this work is to develop a simple yet effective GNN approach, with a focus on empirical analysis to demonstrate scalability across model sizes and data volumes. **This design philosophy follows a trend seen in NLP and CV, where the scalability of GPT and ViT is primarily established through empirical performance rather than theoretical guarantees**. That said, we appreciate the reviewer’s suggestion and agree that incorporating additional theoretical analysis could further strengthen our work. Fortunately, both of the reviewer’s questions can be addressed through a shared lens: by examining the expressive power of random-walk-based representations.
>
> There is already a rich body of theoretical work showing that **random walk-based methods can be more expressive than message-passing neural networks (MPNNs)** under mild assumptions. For example, the GPM [1] (see Theorem 3.4) demonstrates that, under the *reconstruction conjecture*, random walk-based approaches can distinguish all pairs of non-isomorphic graphs, provided a sufficient number of walks are sampled. In essence, random walks capture enough structural information to uniquely characterize a graph, enabling discrimination between graphs that message-passing methods may fail to distinguish. Furthermore, Corollary 3.5 in GPM [1] shows that for any k ≥ 1, there exist graphs that can be distinguished by random walks of length k, but not by the **k-WL isomorphism test**, which bounds the expressive power of many MPNN variants.
>
> Our G²PM framework, which operates on substructures extracted via random walks, can thus inherit this theoretical advantage, **assuming a sufficient number of substructures are sampled**. We believe that these established theoretical frameworks can be directly extended to G²PM with relatively mild assumptions, further emphasizing the expressiveness advantage of G²PM over MPNN.
>
> [1] Beyond Message Passing: Neural Graph Pattern Machine, ICML 25.
>
> > Q2: Several key aspects related to scalability are not thoroughly addressed
> >
>
> Thank you for raising these important points. Before addressing them individually, we would like to clarify that our use of the term **scalability** primarily refers to *scaling laws,* i.e., the ability of the model to benefit from larger datasets and model sizes, as demonstrated in our experiments. Nonetheless, we agree that discussing computational efficiency is important for practical understanding, and we address each sub-question below:
>
> **Q2.1: Memory consumption when applied to large graphs**
>
> **A2.1:** Memory usage is well-managed in G²PM through the use of **mini-batching**, which allows training on large-scale graphs. For instance, given a graph with 2,000 nodes and a mini-batch size of 100, we process only 100 nodes per batch, sampling substructures, encoding them, and computing losses independently. This approach scales efficiently: in our experiments, G²PM handles graphs with over **2 million nodes and 100 million edges** using a single **NVIDIA A40 GPU (48GB)**. We also verified that training can be conducted on a **NVIDIA 3090 (24GB)** by reducing the batch size.
>
> **Q2.2: Computational overhead compared to message-passing baselines**
>
> **A2.2:** We acknowledge that G²PM, being Transformer-based, is computationally more intensive than traditional message-passing methods. For example, on the **ogbn-arxiv** dataset, G²PM with 60M parameters takes approximately **8,000 seconds** for pretraining, while **GraphMAE** (with 3M parameters) takes about **2,000 seconds**.
>
> However, this additional cost is incurred only during the **offline pretraining phase**, which is performed once. In many real-world scenarios, the **learned embeddings are directly reused** for downstream tasks without requiring further inference through the model, making the computational overhead largely irrelevant in practice.
>
> **Q2.3: Impact of walk length on computational cost**
>
> **A2.3:** Walk length directly affects both the **size of substructures** and the computational load. Longer walks generate larger substructures, which increase both memory and compute requirements. However, longer walks do not always yield better performance—excessively long walks may introduce noise and reduce accuracy. The table below shows that **walk length = 8** offers a good trade-off between performance and efficiency.
>
> | **ogbn-arxiv** | **Length=4** | **Length=8** | **Length=16** |
> | --- | --- | --- | --- |
> | **Pretrain Time (s)** | ~6,000 | ~8,000 | ~12,000 |
> | **Accuracy** | 71.4±0.2 | **72.3±0.1** | 72.1±0.2 |
>
> > Q3: How G²PM fundamentally differs from other substructure-based approaches.
> >
>
> In short, G²PM is not just another substructure-based method. It represents a shift in paradigm from enhancing local expressiveness to enabling scalable, generative pretraining over graphs. We provide detailed discussions as follows.
>
> - **A New Perspective:** G²PM introduces a *generative modeling framework* over substructures for graph pretraining, aiming to achieve **scalability in both model size and data volume**. While Transformer-based pretraining and scaling laws are well studied in NLP and CV, this direction remains underexplored in graph learning. In contrast, prior substructure-based methods primarily focus on enhancing model **expressiveness**, specifically addressing the limitations of message-passing in encoding complex substructures. Our approach is orthogonal: it leverages substructures not just for expressiveness but as the foundation for scalable, generative representation learning.
> - **Simple Yet Effective Design:** G²PM offers a simple yet effective implementation of the generative modeling concept. It demonstrates that **strong scalability can be achieved without introducing overly complex architectural components**, in contrast to many prior works relying on complicated model design that fail to achieve scalability.
> - **Scalability in Practice:** Despite its conceptual simplicity, G²PM demonstrates **remarkable scalability**. It successfully scales up to **60M parameters**, significantly outperforming substructure-based or message-passing-based methods like GraphMAE, which tend to saturate at ~3M parameters. This ~20× parameter gap illustrates the practical limitations of existing approaches when applied to larger models and datasets, an area where G²PM excels.
>
> > Q4: Why is the tokenizer fixed in the model? Would a learnable tokenizer further improve performance?
> >
>
> The primary motivation for using a fixed tokenizer is to maintain a **simple yet effective** model design. Despite being fixed, our tokenizer still offers a degree of **adaptability**. The random walk-based substructures are encoded by a Transformer, which can internally learn to focus on informative regions within each walk. In this sense, the model can implicitly identify **sub-substructures** within each sampled sequence, enabling meaningful representation learning even without a learnable tokenizer.
>
> That said, we agree that **learnable or adaptive tokenizers** represent a promising direction for future improvement. Such approaches could tailor the substructure sampling process to better suit different graph domains. For example, one could employ **reinforcement learning** to adapt the p and q parameters of the random walk strategy based on the graph type—favoring short, localized walks (small p, large q) for social networks, and longer-range walks (large p, small q) for molecular graphs.
>
> > Q5: What make G²PM practical on billion-edge graphs?
> >
>
> In short, we use mini-batching to enable the model to scale to billion-edge graphs. Due to the space limitation, please refer to Q2.1 for details.
>
> > Q6: How does G²PM address scenarios where key higher-order patterns aren't sufficiently captured by walk-based sampling?
> >
>
>
> G²PM addresses this by **increasing the walk length**, which improves the likelihood of covering complex patterns. For instance, on the PCBA molecular dataset, using a walk length of 16, G²PM achieves **75.6 AUC**, outperforming the best generative baseline (73.2 AUC), indicating its ability to approximate higher-order structures.
>
> That said, random walks cannot guarantee full coverage of complex patterns. While using a **predefined vocabulary** of substructures could improve coverage, it introduces high computational cost and requires domain expertise. In contrast, our random walk-based approach is **efficient, domain-agnostic**, and scalable, offering a practical trade-off between expressiveness and efficiency.
>
> > Q7: Could you provide more insight into why positional embeddings show no benefit in your framework?
> >
>
> G²PM treats substructures, not individual nodes, as tokens. The model’s goal is to identify and learn from informative patterns, rather than relying on the exact positions of nodes in the graph. In this setting, global node positions become less relevant, as the focus shifts to the semantic contribution of substructures. As a result, incorporating node-level positional embeddings has minimal impact on performance within our framework.

---

> > ### Comment · Reviewer_85SU · 2025-08-07
> >
> > Thank you for thoughtfully addressing all my questions and concerns in your rebuttal. I truly appreciate the effort and clarity you've demonstrated, which has led me to increase my score.

---

> > > ### Author Response · Authors · 2025-08-07
> > >
> > > Thank you for your kind words. We truly appreciate your insightful suggestions, which help us further enhance the quality of our work!

---

### Official Review · Reviewer_XK3h · 2025-07-02

**Clarity:** 3
**Significance:** 3
**Originality:** 3
**Rating:** 4
**Confidence:** 4

**Summary:**

This paper introduces G²PM, a generative pre-training framework for graph representation learning based on Transformers. The key idea is to model graphs as sequences of substructures and apply Masked Substructure Modeling (MSM), where masked substructures are reconstructed conditionally on visible ones. The proposed approach enables the application of autoregressive transformer models to graph data by leveraging substructure-aware sequential representations.

**Questions:**

**Scope of downstream tasks**: The current architecture removes the decoder and adds a linear prediction head to the encoder. This setup supports node-level and graph-level prediction but not generation. Is it possible to extend this framework to graph generation tasks?

**Ethical Concerns:**

["NO or VERY MINOR ethics concerns only"]

**Final Justification:**

The rebuttal solved my minor concerns and I keep my positive scores.

**Limitations:**

The paper only discusses the future direction and does not include any potential negative societal impact.

**Quality:**

3

**Strengths And Weaknesses:**

[Strenghts]

- **Clear and well-motivated contribution**: The paper is grounded in a well-defined need—the incorporation of semantic granularity into graph representations—to improve the effectiveness of Transformer-based models on graphs. The integration of substructures into a sequential modeling framework is conceptually novel.
- **Thorough experimentation**: The authors provide comprehensive empirical results across various benchmarks, demonstrating the effectiveness of the proposed method.

[Weaknesses]

- **Missing prior work on graph-to-sequence representations**: The manuscript overlooks several prior works [1–5] that represent graphs as sequences suitable for Transformer-style training. Notably, the random walk approach employed in this paper closely resembles the method proposed in [5]. While the scope of this work focuses on pre-training, these related methods should still be acknowledged in the “Graph Transformers” and “Random Walks on Graphs” sections of the Related Work.
- **Representation invariance and completeness**: The sequential representation is based on sampling and random walks. It is unclear whether the resulting representation is lossless. Specifically, how does the model capture the attachment points between substructures? A discussion on the limitations or assumptions of the representation would improve transparency.
- **Limited benefit of cross-domain pretraining**: Although the paper aims to establish a graph pre-training framework, cross-domain pre-training occasionally degrades performance. For example, in Table 5, the performance on the HIV dataset is worse than in Table 3. The authors should elaborate on potential causes (e.g., domain mismatch or lack of generalization) and whether this highlights a limitation of their pre-training framework.

---

> ### Author Rebuttal · Authors · 2025-07-30
>
> Dear reviewer,
>
> We sincerely thank the reviewer for the thoughtful and constructive feedback. We are pleased that the core contributions of G²PM—particularly the integration of substructure-aware sequential modeling for Transformer-based graph pretraining—were found to be clear, well-motivated, and empirically validated. We appreciate the recognition of the novelty in using substructures for masked modeling, as well as the thoroughness of our experimental evaluation. At the same time, we acknowledge the helpful suggestions regarding missing related work on graph-to-sequence representations, the need for more discussion on representation completeness and attachment semantics, and clarification on the cross-domain transfer results. In this rebuttal, we aim to address these concerns and further clarify the scope, assumptions, and extensibility of our framework.
>
> > Q1: Missing prior work on graph-to-sequence representations
> >
>
> Thank you for pointing this out. We apologize for the oversight and appreciate the reminder to more thoroughly position our work within the graph-to-sequence literature. While the reviewer did not specify the references, we will make a dedicated effort to identify and cite relevant works [1-5] that model graphs as sequences, particularly those leveraging **random walks or sequential encoding for Transformer-style models**. We will ensure these are properly discussed in the **“Graph Transformers”** and **“Random Walks on Graphs”** sections of the Related Work in the revised version of the paper.
>
> Additionally, we will clarify how **G²PM fundamentally differs from prior sequence-based methods**. While we acknowledge that there are works exploring graph-to-sequence representations, most of these methods are **task-specific** and do not focus on building **scalable, general-purpose pretraining frameworks**. For example, many existing approaches (i) rely on node-level tokenization, which limits their capacity to capture rich substructure semantics, or (ii) are primarily designed for small-scale or domain-specific tasks (e.g., molecule generation) without addressing scalability to large graphs and datasets.
>
> In contrast, our focus is on **masked substructure modeling** as a **generative pretraining paradigm** that scales effectively with both **model size (up to 60M parameters)** and **data volume**, as well as enables strong cross-domain performance. Unlike prior sequence-based methods that often lack scalability or fail to fully leverage substructure-level granularity, G²PM introduces a **substructure-as-token design** that allows Transformers to learn transferable representations across diverse graph types. We will make these distinctions explicit in the revised Related Work section to better highlight the unique contributions of our approach.
>
> [1] Graph Generative Pre-trained Transformer, ICML 25
>
> [2] GraphGPT: Generative Pre-trained Graph Eulerian Transformer, ICML 25
>
> [3] Graph-Mamba: Towards Long-Range Graph Sequence Modeling with Selective State Spaces, arXiv 24
>
> [4] Graph Mamba: Towards Learning on Graphs with State Space Models, KDD 24
>
> [5] Supercharging Graph Transformers with Advective Diffusion, ICML 25
>
> > Q2: Representation invariance and completeness
> >
>
> Thank you for raising this important point. Our current design does not aim to achieve strict **invariance** or **completeness** in the traditional sense. Instead, we intentionally rely on **randomized sampling** to allow the model to learn which patterns are important and which are not. The randomness ensures **diverse and comprehensive substructure coverage**, avoiding reliance on predefined heuristics or structural assumptions that may introduce inductive bias and limit generalization.
>
> That said, our method can be readily extended to incorporate sequential consistency and potentially improve completeness. For example, instead of independently sampling 50 random walks of length 8, one could generate a **single long walk of length 400** and split it into 50 segments. This approach would naturally preserve **sequential continuity** between substructures and could help retain attachment context across sampled patterns.
>
> We acknowledge that our current approach may not fully capture all attachment relationships between substructures, and we will include a discussion of this limitation—and possible extensions—in the revised **Limitations** section of the paper.
>
> > Q3: Limited benefit of cross-domain pretraining
> >
>
> Thank you for raising this concern. We believe there may be a misunderstanding regarding the datasets referenced in Table 3 and Table 5. Although both refer to **HIV**, they are **different versions** of the dataset.
>
> - In **Table 3**, the HIV dataset is the **standard version** that includes **atom properties as features**.
> - In **Table 5**, the HIV dataset includes **textual node attributes**, where nodes are annotated with **textual descriptions** and features are derived from a **text encoder**.
>
> Therefore, there is naturally a performance gap between these two tables. For a fair comparison, we adopt the dataset used in Table 5 to evaluate whether the cross-domain pretraining can enhance model performance. In particular, we pretrain G²PM using **only the HIV dataset** and the model achieves **72.1 AUC**. In contrast, when we apply **cross-domain pretraining** using three datasets (in Table 5), the performance improves to **74.1 AUC**. This clearly demonstrates that cross-domain pretraining **does enhance model performance**, even under domain shift. We will clarify this distinction in the paper to avoid confusion.
>
> > Q4: Is it possible to extend this framework to graph generation tasks?
> >
>
> Yes, our framework can be naturally extended to support graph generation, and we envision two promising directions:
>
> 1. **Autoencoder-style generation:** G²PM can serve as a powerful **feature encoder** within a standard graph generation pipeline, such as graph autoencoders or diffusion-based models. A straightforward approach is to feed a graph into G²PM to obtain a latent graph representation, and then use a **decoder module** (e.g., GNN or MLP) to reconstruct the graph structure.
> 2. **Autoregressive generation:** A more exciting extension is to adapt G²PM for **autoregressive generation**. Instead of using random walks, we could apply traversal strategies like **DFS or BFS** to generate a long sequence of nodes, which is then split into substructure segments. The model is trained to **predict the next segment** conditioned on the previous ones, effectively learning to reconstruct the original traversal sequence. Since traversal sequences (e.g., DFS/BFS) contain sufficient structural information to reconstruct the original graph, this approach allows G²PM to be naturally adapted for **graph generation tasks**.
>
> Both directions align well with the substructure-based design of G²PM and open the door for future work on generative modeling. We appreciate the reviewer for suggesting this valuable extension.

---

> > ### Comment · Reviewer_XK3h · 2025-08-06
> >
> > Sorry for the missing reference in the review. I acknowledged it just right before and the reference I mentioned are as below:
> >
> > [1] A simple and scalable representation for graph generation. Jang et al., ICLR 2024
> >
> > [2] Graph generation with K2-trees. Jang et al., ICLR 2024
> >
> > [3] Graphgen: A scalable approach to domain-agnostic labeled graph generation. Goyal et al, Proceedings of the web conference, 2020
> >
> > [4]  Pure Transformers are Powerful Graph Learners. Kim et al., NeurIPS 2022.
> >
> > [5] Revisiting random walks for learning on graphs. Kim et al., ICLR 2025.
> >
> > It would be wonderful if the future manuscript integrates not only them but also the references mentioned by the authors.
> > Thank you for the detailed response and as my raised weaknesses are minor, I keep my positive scores.

---

> > > ### Author Response · Authors · 2025-08-06
> > >
> > > Thank you for your valuable suggestions. We'd like to include these papers in our revised manuscript!

---

### Official Review · Reviewer_CueM · 2025-07-02

**Clarity:** 4
**Significance:** 3
**Originality:** 1
**Rating:** 5
**Confidence:** 2

**Summary:**

The authors propose a Transformer-based pre-training framework for graphs based on masked subgraph prediction. Subgraphs are generated through random walk tokenization and the model is shown to perform well across a range of scaling and ablation experiments.

**Questions:**

As well as addressing the two points in weaknesses, I have a couple of minor questions.

[Line 128] How do you decide the ordering of the graph substructure tokens? Does this matter?

Do the substructures from random walks significantly differ across datasets? Would this effect cross-domain adaption?

**Ethical Concerns:**

["NO or VERY MINOR ethics concerns only"]

**Final Justification:**

The authors have satisfactorily addressed my concerns in their rebuttal.

**Limitations:**

yes

**Quality:**

3

**Strengths And Weaknesses:**

Strengths

- The authors undertake a comprehensive set of ablations and experiments. These clearly highlight the benefits of using subgraphs generated from their tokenization with a transformer encoder across a range of tasks.

- The cross-domain transfer experiments are very compelling and show how the approach could make sense as a form of foundation model when compare to standard message-passing based approaches.

- The paper is well-written and clearly motivated.

Weaknesses

- the technical development is limited in that the model is based on GPM and the innovation comes from the masked modeling component which in itself is well established. This lack of innovation is further highlighted by the fact that the performance improvement over GPM is negligible across a lot of tasks and can reduce performance. Given that the paper mainly adapts standard self-supervised learning techniques to GPM and doesn't see a huge performance boost, the scope of the paper could appear limited.

- The self-supervised step can introduce a lot of additional computational time. Having some idea of the tradeoff between this and the performance improvement could be helpful to practitioners.

---

> ### Author Rebuttal · Authors · 2025-07-30
>
> Dear reviewer,
>
> We sincerely thank the reviewer for the thoughtful and constructive feedback. We are pleased to hear that the paper is considered well-written, clearly motivated, and supported by a comprehensive set of experiments and cross-domain transfer evaluations. We appreciate the recognition of G²PM’s potential as a foundation model alternative to message-passing-based approaches. We aim to address your concerns by clarifying the contributions beyond prior work, discussing the practical trade-offs of our method, and providing insights into the ordering and variability of substructure tokenization across datasets.
>
> > Q1: The technical development is limited. The improvements over GPM is limited.
> >
>
> Thank you for your thoughtful feedback. We believe there may be a misalignment in understanding, and we would like to clarify our contributions on two fronts: (1) the novelty and significance of the G²PM framework, and (2) the performance improvement of G²PM over GPM.
>
> **1. Clarifying the Novelty and Contribution of G²PM:**
>
> - **Scalable Transformer-based Graph Pretraining:** G²PM explores the underexplored territory of scalability in graph representation learning using Transformer-based generative pretraining. While Transformer scalability has been extensively studied in NLP and CV, there is limited work demonstrating effective and scalable Transformer architectures for graphs. Our method is, **to the best of our knowledge, the first to achieve such scalability**, scaling up to 60M parameters, whereas representative graph pretraining methods like GraphMAE saturate around 3M parameters, showing a **~20× benefit**.
> - **Simple yet Effective:** G²PM achieves strong performance with a minimal and straightforward design, **avoiding unnecessary architectural complexity while still being extensible**. As shown in our ablation studies, adding components such as positional encodings or topology-aware reconstruction further improves performance, indicating the method’s robustness and flexibility.
> - **Design Insights for Scalable Graph Models:** We conduct an extensive design space exploration, **offering practical insights into building scalable graph foundation models**. G²PM shows promise as a potential backbone model due to (1) its ability to learn from diverse graphs at scale, and (2) its substructure-based tokenization, which enables effective cross-domain transfer by capturing reusable graph motifs.
>
> **2. Clarifying Performance Improvements over GPM:**
>
> We also want to address the concern regarding the seemingly marginal performance gain of G²PM over GPM. This is primarily due to the difference in training paradigms:
>
> - **GPM** is a fully supervised model, trained end-to-end with label supervision, allowing all parameters to be updated during training.
> - **G²PM**, on the other hand, is trained in a self-supervised fashion using masked substructure prediction. In our reported setup, we freeze the pretrained model and only train a **linear classifier** on top of the learned representations.
>
> This discrepancy in training protocols naturally leads to a performance gap. However, when we **fine-tune the entire G²PM model** on downstream tasks (rather than only training the classifier), we observe **significant performance gains**, surpassing GPM across both node and graph classification tasks, as shown below. These results indicate that G²PM offers more than marginal gains—it can outperform strong supervised baselines when properly fine-tuned. We will clarify this more explicitly in the final version of the paper.
>
> | **Method** | **Arxiv** | **Products** | **PCBA** | **REDDIT-M12K** |
> | --- | --- | --- | --- | --- |
> | **Graph Type** | Citation | E-Commerce | Molecule | Social |
> | **# Graphs** | 1 | 1 | 437,929 | 11,929 |
> | **# Nodes** | 169,343 | 2,449,029 | ∼26.0 | ∼391.4 |
> | **# Edges** | 2,315,598 | 123,718,024 | ~28.1 | ∼913.8 |
> | **GPM** | 72.9±0.7 | 82.6±0.4 | 75.1±0.3 | 43.1±0.3 |
> | **G2PM - Linear Classifier (as reported)** | 72.3±0.1 | 80.6±0.0 | 75.6±0.1 | 41.8±0.3 |
> | **G2PM - Full Fine-Tuning** | **73.5±0.4** | **83.1±0.3** | **76.8±0.2** | **43.8±0.3** |
>
> > Q2: The self-supervised step can introduce a lot of additional computational time. Having some idea of the tradeoff between this and the performance improvement could be helpful to practitioners.
> >
>
> Thank you for raising this practical concern. We agree that self-supervised learning introduces additional computational cost. However, we would like to emphasize that in real-world applications, this cost is typically incurred during an **offline pretraining phase**. Once the model is pretrained, it can be deployed for downstream tasks with **no additional overhead**, as the learned node or graph embeddings can be directly used during inference. Therefore, training efficiency is less of a concern in many industrial settings where inference-time speed is prioritized.
>
> That said, we also provide empirical data to help practitioners understand the trade-off between computation time and performance gain. As shown in the table below, pretraining with G²PM results in **substantial improvements** across all datasets. Even on large-scale graphs with over 2 million nodes (e.g., ogbn-products), the total pretraining time is **less than 24 hours** on a single NVIDIA A40 GPU—an acceptable cost for many production environments given the performance benefits.
>
> | Dataset | **# Nodes / Graphs** | **G²PM (w/o Pretrain)** | **G²PM (with Pretrain)** | Imp. (%) | **Pretrain Time (s)** | **Inference Time (s)** |
> | --- | --- | --- | --- | --- | --- | --- |
> | **Arxiv** | 169,343 nodes | 69.6±0.1 | 72.3±0.1 | **3.88** | ~8,000 | ~190 |
> | **Products** | 2,449,029 nodes | 76.9±0.2 | 80.6±0.0 | **4.81** | ~60,000 | ~5,300 |
> | **PCBA** | 437,929 graphs | 68.4±0.0 | 75.6±0.1 | **10.53** | ~45,000 | ~8,700 |
> | **REDDIT-M12K** | 11,929 graphs | 37.5±0.3 | 41.8±0.3 | **11.47** | ~1,000 | ~60 |
>
> > Q3: How do you decide the ordering of the graph substructure tokens? Does this matter?
> >
>
> In our current approach, we do **not** explicitly define an order for the substructure tokens. Since we adopt a **masked-token prediction** strategy for pretraining, the method is inherently **order-invariant**—the learning objective and model architecture are not sensitive to the order of tokens.
>
> However, we acknowledge that token ordering becomes important in **next-token prediction** frameworks, which we plan to explore in future work. In such cases, defining a meaningful and consistent order over graph substructures will be essential. One intuitive approach could be to use the **pattern size** (e.g., the number of unique nodes) as an ordering heuristic—predicting larger patterns from smaller ones. Alternatively, other indicators such as **graph edit distance** or **spectral properties** could be employed to define a semantically meaningful order.
>
> We appreciate the reviewer bringing attention to this point, which is an important direction for future research.
>
> > Q4: Do the substructures from random walks significantly differ across datasets? Would this effect cross-domain adaption?
> >
>
> Yes, the substructures generated via random walks can differ across datasets, and this difference can indeed impact cross-domain adaptation performance. The effectiveness of transfer learning largely depends on the model's ability to **capture invariant and semantically meaningful substructures** that are shared across domains.
>
> For example, certain patterns, like high-degree nodes, may carry similar semantic meanings in both social and citation networks (e.g., representing influential users or highly cited papers). These shared substructures facilitate successful adaptation. However, when the source and target graphs come from very different domains, they are **less likely to share similar substructures**, which reduces the overlap in learned representations and can lead to diminished transfer performance.
>
> **Despite this challenge, our method shows promising results in cross-domain settings**, suggesting that some level of generalizable substructure information is still being captured. Further improving the alignment of substructure semantics across domains remains an exciting direction for future research.

---

> ### Comment · Reviewer_CueM · 2025-08-03
> **response**
>
> Thank you for the response - my concerns have been addressed

---

> > ### Author Response · Authors · 2025-08-03
> >
> > Thank you for your insightful suggestions! Your comments are really helpful for our work.

---

### Official Review · Reviewer_P8WG · 2025-07-03

**Clarity:** 3
**Significance:** 3
**Originality:** 3
**Rating:** 4
**Confidence:** 4

**Summary:**

The authors claim that existing pretrain graph representation methods suffer from fundamental limitations due to MPNNs: constrained expressive power, over smoothing, over-squashing, and poor capacity for modeling long-range dependencies. Therefore, this paper aims to extend the success of Transformer-based generative pre-training to the graph domain, with the goal of enabling scalable graph representation learning. The authors propose G^{2}PM, which first applies random walk to extract node sequence for each node, and then obtain the nodes’ representation via a transformer. Finally, based on constructed substructures sequence, mask-prediction self-supervised loss is used to learn node representations. Experimental results on multiple datasets seem to validate the effectiveness of the pre-training.

**Questions:**

See the weaknesses.

**Ethical Concerns:**

["NO or VERY MINOR ethics concerns only"]

**Final Justification:**

The authors have addressed my concerns, so I decide to increase my score.

**Limitations:**

The authors claim in the limitation section that the current method is merely a masked-token prediction paradigm, and its effectiveness and scalability for next-token prediction remain unknown. Exploring next-token prediction is particularly important, as autoregressive methods have shown greater potential than autoencoding methods in the fields of NLP and CV for building foundation models.

**Quality:**

2

**Strengths And Weaknesses:**

Strengths:
1. The paper is easy to read and understand.
2. The experiment evolve node classification from 7 datasets and graph classification from 7 datasets.
3. The proposed G^{2}PM seems to have somewhat scalability from Figures 2 and 3, which is attractive.
4. The paper conducts lots of experimental analysis about model architecture design in Section 3.
Weakness:
1. The proposed method is not a novelty. According to my understanding, the proposed G²PM obtains the substructure sequence of each node through random walks and then computes the node representation using sequence modeling methods (i.e., Transformer, RNN). Based on the constructed node representations and the sampled sequences, G²PM further learns the node representations using a general mask prediction method. Thus, the main innovation of the method is tokenizing substructures via random walks, which is simple and straightforward.
2. In the scalability experiment of Figure 2, 60M is not a large model size. If the model size continues to increase, can the performance of the method still improve consistently?
3. In the experiment of Model architecture, the author claims that deeper encoders enhance performance while increasing decoder capacity provides limited or negative returns. I personally think that the selected range of encoder and decoder layers is too small to support the argument. Moreover, which dataset is used for the experimental results in Figure 5(b)?
4. To demonstrate the pretrain effectiveness, it is necessary to experiment with a link prediction task.
5.“Model Layer. Table 5a shows” in line 181 has a linking error.

---

> ### Author Rebuttal · Authors · 2025-07-30
>
> Dear reviewer,
>
> We sincerely thank the reviewer for the thoughtful and detailed feedback. We appreciate the recognition of the paper’s clarity, the comprehensive experimental analysis, and the promising scalability of our proposed method. We aim to clarify our contributions, provide additional experimental evidence, and address the limitations discussed to better convey the value and potential of G²PM.
>
> > Q1: The method is simple and straightforward.
> >
>
> Thank you for your valuable feedback. We appreciate the opportunity to clarify the novelty of our method and address your concerns.
>
> 1. **A New Perspective on Graph Pretraining:**
>
>     Our primary contribution lies in introducing a new and promising *generative modeling perspective* over substructures for graph pretraining, which enables substantial scalability. While the connection between Transformer-based pretraining and scalability has been extensively explored in other domains such as NLP and CV, this direction remains underexplored in the context of graph learning. To the best of our knowledge, **no prior work has successfully demonstrated such scalability in graph pretraining**.
>
> 2. **Simple Yet Effective Design:**
>
>     Our method proposes a simple yet effective implementation of this generative framework. While the use of random walks for substructure sampling is straightforward, our results demonstrate that even such a **simple tokenization can yield strong performance** when combined with a generative modeling approach. Importantly, this simplicity does not limit extensibility. As evidenced in Tables 1(b) and 1(e), incorporating advanced components such as positional encodings and topological reconstruction can further boost performance, indicating that G²PM offers a strong and adaptable foundation.
>
> 3. **Scalability Advantage:**
>
>     Despite its conceptual simplicity, G²PM achieves compelling scalability with respect to both model capacity and data volume. In contrast to existing message-passing-based pretraining methods, which often involve complex architectural designs yet struggle to scale, G²PM supports model scaling up to 60M parameters. However, GraphMAE, a widely adopted method, saturates at around 3M parameters, **highlighting a significant (~20x) scalability advantage of G²PM**.
>
>
> In summary, we believe that G²PM not only offers a novel and scalable approach to graph pretraining, but also serves as a potential backbone for the next generation of graph representation learning, moving beyond the limitations of traditional message passing.
>
> > Q2: In the scalability experiment of Figure 2, 60M is not a large model size. If the model size continues to increase, can the performance of the method still improve consistently?
> >
>
> **1. Model Size in Graph Learning:**
>
> While 60M parameters may not appear large compared to models in the LLM era, where even the smallest models typically exceed 1B parameters, it is **indeed a substantial size in graph neural networks**. Graph models generally operate at a much smaller scale. For example, GraphMAE, a representative graph pretraining method, has approximately 3M parameters under its default configuration. Even among larger graph foundation models, parameter counts rarely exceed 30M unless they explicitly incorporate LLMs (see Table 8 in [1]). Therefore, achieving 60M parameters in a standalone graph model is already a significant step forward in scalability for this domain.
>
> **2. Performance Scaling with Model Size:**
>
> The impact of increasing model size on performance depends heavily on the scale of the dataset. On smaller or medium-sized datasets such as *ogbn-arxiv* (~170K nodes), increasing the parameter count beyond a certain point leads to saturation or even slight performance drops due to overparameterization. For example, as shown in the table below, performance peaks around 60M and then declines slightly with further increases.
>
> However, on large-scale datasets like *ogbn-products* (~2M nodes), **increasing model size continues to yield consistent improvements**. As the table shows, **G²PM scales up to 160M parameters and continues to improve in performance**. We emphasize that this does **not** imply that G²PM has reached its performance ceiling; we have not tested models beyond 160M due to the time constraints of the rebuttal period.
>
> | **Dataset** | **~60M** | **~120M** | **~160M** |
> | --- | --- | --- | --- |
> | **ogbn-arxiv** | 72.31±0.07 | 71.42±0.09 | 71.41±0.14 |
> | **ogbn-products** | 80.56±0.01 | 80.98±0.02 | 81.31±0.01 |
>
> [1] GFT: Graph Foundation Model with Transferable Tree Vocabulary, NeurIPS 24.
>
> > Q3: In the experiment of Model architecture, the author claims that deeper encoders enhance performance while increasing decoder capacity provides limited or negative returns. I personally think that the selected range of encoder and decoder layers is too small to support the argument.
> >
>
> As previously mentioned, model performance on **ogbn-arxiv** tends to saturate beyond 60M parameters. Therefore, to better investigate the effect of encoder and decoder depth, we conducted additional experiments on the larger **ogbn-products** dataset, where model scaling exhibits clearer trends.
>
> From these results, we observe a **consistent pattern**: increasing the encoder depth leads to improved performance, while increasing decoder depth beyond a certain point yields diminishing or even negative results. Specifically, a shallow decoder (e.g., 1 layer) paired with a deeper encoder results in the strongest performance. **This supports our original statement regarding the asymmetric benefit of deeper encoders versus decoders**.
>
> Due to computational constraints, we extended encoder depth up to 6 layers and decoder depth up to 3 layers. We would be happy to include additional results if required.
>
> | **Enc. Layers** | **Dec. Layers** | **ogbn-products** |
> | --- | --- | --- |
> | 1 | 1 | 78.39±0.04 |
> | 2 | 1 | 79.61±0.02 |
> | 3 | 1 | 80.56±0.01 |
> | 6 | 1 | 81.04±0.01 |
> | 6 | 2 | 80.67±0.03 |
> | 6 | 3 | 80.13±0.01 |
>
> > Q4: Link Prediction
> >
>
> Thank you for the suggestion. To further demonstrate the effectiveness of our pretraining approach, we provide link prediction results on three datasets: *Cora* (2,708 nodes, 10,556 edges), *Pubmed* (19,717 nodes, 88,648 edges), and *ogbl-collab* (235,868 nodes, 2,570,930 edges). We follow a standard 80/5/15 split for training, validation, and test sets. For evaluation, we report *Hit@20* on Cora and Pubmed, and *Hit@50* on ogbl-collab. The results are summarized in the table below, where we compare our method (G²PM) with a basic supervised GCN and a widely used graph pretraining method, GraphMAE. As shown, GraphMAE significantly outperforms GCN, benefiting from its ability to utilize unlabeled data during pretraining. Notably, G²PM achieves the best performance across all datasets, likely due to its capacity to model substructures that capture latent connectivity patterns between nodes.
>
> | **Method** | **Cora (Hit@20)** | **Pubmed (Hit@20)** | **ogbl-collab (Hit@50)** |
> | --- | --- | --- | --- |
> | **GCN** | 84.14±1.19 | 85.06±3.79 | 44.75±1.07 |
> | **GraphMAE** | 87.64±1.44 | 87.12±3.15 | 46.53±0.89 |
> | **G2PM** | **90.42±0.78** | **88.01±3.79** | **47.10±0.57** |
>
> > Q5: Can we replace masked-token prediction with next-token prediction?
> >
> 1. **Next-token prediction may not necessarily outperform masked-token prediction in G²PM:**
>
>     While autoregressive (next-token) prediction has shown strong performance in sequential domains like text, its advantages may not directly transfer to graph-structured data. For example, in image generation, many autoregressive methods underperform compared to denoising-based approaches like diffusion models. **This is because images—and similarly, graphs—often contain high levels of redundancy or noise.** Denoising methods like masked-token prediction can learn to identify and reconstruct informative regions, discarding unimportant patterns.
>
>     In our setting, G²PM benefits from this principle. For instance, in social networks, detecting key substructures such as triangles may suffice for node classification, while other sampled patterns may act as noise. From this perspective, masked-token prediction is a natural fit for our model, and next-token prediction may not offer clear benefits in terms of scalability or effectiveness.
>
> 2. **Extending G²PM to next-token prediction is feasible and worth exploring:**
>
>     We agree that exploring next-token prediction is important, particularly for advancing generative graph modeling. The main challenge lies in defining a meaningful order over graph substructures, which is non-trivial due to the non-Euclidean nature of graphs. Nevertheless, there are several promising directions: (1) **Pattern size**: use the number of unique nodes in a substructure to define an order (e.g., predict larger patterns from smaller ones). (2) **Graph edit distance**: order substructures based on how dissimilar they are from one another. (3) **Spectral properties**: use eigenvalue-based metrics to define substructure complexity and ordering. We acknowledge this as a valuable direction for future work and appreciate the reviewer’s suggestion in pointing it out.
>
>
> > Q6: Minor points
> >
>
> Q6.1: which dataset is used for the experimental results in Figure 5(b)?
>
> A6.1: The dataset used in Figure 5(b) is **ogbn-arxiv**. Thank you for pointing this out—we will clarify this explicitly in the revised manuscript.
>
> Q6.2: “Model Layer. Table 5a shows” in line 181 has a linking error.
>
> A6.2: Thank you for catching this. We will correct the linking error in the updated version of the paper.

---

### Comment · Area_Chair_mEiz · 2025-08-05
**Please participate in the discussions and respond to the authors**

Dear Reviewers,

Thank you for your valuable reviews. With the Reviewer-Author Discussions deadline approaching, please take a moment to read the authors' rebuttal and the other reviewers' feedback, and participate in the discussions and respond to the authors. Finally, be sure to complete the "Final Justification" text box and update your "Rating" as needed. Your contribution is greatly appreciated.

Thanks.\
AC

---

### Decision · Program_Chairs · 2025-09-17

**Decision:**

Accept (poster)

**Comment:**

Summary:
This paper addresses the challenge of scalable graph generative modeling by introducing the Generative Graph Pattern Machine, a pre-training framework that adapts the Transformer architecture for graphs. This approach represents graph instances—including nodes, edges, and entire graphs—as sequences of substructures, then uses generative pre-training over these sequences to learn transferable, generalizable representations. Experiments on multiple datasets demonstrate the effectiveness of the proposed model.

Strengths:
1. The paper is well-structured and easy to follow.
2. The authors conduct a comprehensive set of ablations and experiments.
3. The work is grounded in a clear need to improve the effectiveness of Transformer-based models on graphs. The integration of substructures into a sequential modeling framework is a conceptually novel idea.


Weaknesses:
1. Some experimental aspects could be improved. For instance, the scalability experiment in Figure 2 would benefit from evaluating larger model sizes. Including a link prediction task would also be valuable, as it is a widely considered task.
2. The paper should discuss the trade-off between the cost of the self-supervised pre-training step and the resulting performance improvements.
3. A more thorough discussion of prior work on graph-to-sequence representations is required to strengthen the related work section.


In summury, this paper propose a novel pretraining framework on graph learning, which is different from the most utilized message-passing paradigm. New paradigms are driving force to the graph learing area. The experiments also show the effectiveness of the proposed model. I recommend the authors to address the minor issues in the camera ready version.